# A National Catalogue of Viruses Associated with Indigenous Species Reveals High-Throughput Sequencing as a Driver of Indigenous Virus Discovery

**DOI:** 10.3390/v14112477

**Published:** 2022-11-09

**Authors:** Merlyn Robson, Kar Mun Chooi, Arnaud Gérard Blouin, Sarah Knight, Robin Marion MacDiarmid

**Affiliations:** 1School of Biological Sciences, University of Auckland, Private Bag 92019, Auckland Mail Centre, Auckland 1142, New Zealand; 2Bio-Protection Research Centre, Lincoln University, P.O. Box 85084, Lincoln 7674, New Zealand; 3The New Zealand Institute for Plant and Food Research Limited, Private Bag 92169, Auckland 1142, New Zealand; 4Virology-Phytoplasmology Laboratory, Agroscope, 1260 Nyon, Switzerland

**Keywords:** virus, Aotearoa New Zealand, host, indigenous, exotic, endemic, high-throughput sequencing, symptom, non-targeted, targeted

## Abstract

Viruses are important constituents of ecosystems, with the capacity to alter host phenotype and performance. However, virus discovery cued by disease symptoms overlooks latent or beneficial viruses, which are best detected using targeted virus detection or discovered by non-targeted methods, e.g., high-throughput sequencing (HTS). To date, in 64 publications, 701 viruses have been described associated with indigenous species of Aotearoa New Zealand. Viruses were identified in indigenous birds (189 viruses), bats (13 viruses), starfish (4 viruses), insects (280 viruses), and plants (126 viruses). HTS gave rise to a 21.9-fold increase in virus discovery rate over the targeted methods, and 72.7-fold over symptom-based methods. The average number of viruses reported per publication has also increased proportionally over time. The use of HTS has driven the described national virome recently by 549 new-to-science viruses; all are indigenous. This report represents the first catalogue of viruses associated with indigenous species of a country. We provide evidence that the application of HTS to samples of Aotearoa New Zealand’s unique fauna and flora has driven indigenous virus discovery, a key step in the process to understand the role of viruses in the biological diversity and ecology of the land, sea, and air environments of a country.

## 1. Introduction

Understanding what viruses are present in an ecosystem, e.g., an island nation, is important for the protection of both indigenous and introduced species, providing important security economically, socially, culturally, and/or environmentally. The extant virome is important knowledge for the islands of Aotearoa New Zealand due to the unique status of a flora and fauna increasingly at risk from human activity. Cataloguing the extant viruses associated with indigenous species can allow for the development of control measures against those that cause diseases, and provide a greater understanding of those that could influence population and host ecosystems. Such catalogues are periodically published for the viruses associated with the plants (mainly exotic crop plants) within Aotearoa New Zealand, but are absent for other virus host types [1,2,3].

Viruses are most commonly known and perceived through the diseases they cause. In the past few years, the study and discovery of viruses have become easier and more common, with the development of advanced molecular tools allowing their untargeted detection [4,5,6,7,8,9]. These new approaches triggered the discovery of viruses from symptomless, healthy-looking organisms, suggesting that the majority of viruses are latent or may be beneficial to their immediate host, or within higher order relationships, e.g., tripartite symbioses, whereby a virus may benefit both its endophytic fungal host and the endophyte’s plant host [10]. 

Aotearoa New Zealand presents a unique place of study for many species, including viruses. Owing to its relatively recent discovery and settlement (by Māori, the indigenous people of Aotearoa New Zealand), and subsequent colonization (by British and Europeans and other peoples from around the world), there has been less time for anthropogenic influence than in many other places worldwide [11,12,13]. The relative lack of anthropogenic influence is especially evident on some offshore islands, where humans either did not settle or settlement was brief [14]. Aotearoa New Zealand’s biogeographic history also provides a unique subset of evolved flora and fauna, as its geographical isolation has enabled the development of separate evolutionary lineages that are now endemic to the country [15]. This unique geographical isolation provides the opportunity for greater understanding of the impacts of viruses on a country’s overall ecosystem dynamics. 

This meta-analysis examines previously described viruses associated with Aotearoa New Zealand indigenous species. It reviews how the research and discovery of these viruses occurred, how a virus could be defined as indigenous, and the change in methods used for viral discovery and their influences. We also highlight aspects such as viral abundances by family, the potential impacts and management of viruses in indigenous species, and current and future methods of viral detection.

## 2. Materials and Methods

### 2.1. Catalogued Viruses

The criterion for viruses to be included in this meta-analysis is a virus associated with one or more species that is indigenous to Aotearoa New Zealand and has been described in a peer-reviewed journal publication. 

Viruses associated with Aotearoa New Zealand indigenous species were compiled through multiple methods. The keywords (viruses, New Zealand, Aotearoa, plants, fungi, fish, insects, birds, bats, bacteria, archaea, oomycetes, indigenous, exotic) were searched across Google, Google Scholar, and PubMed databases between the dates September 2021 to February 2022. Between December 2021 and February 2022, PubMed and Google Scholar databases were searched for relevant publications from key authors, including A. Varsani, S. Kraberger, P. Guy, C. Delmiglio, and M. N. Pearson. Fourteen virologist or key contacts were personally asked to provide information about, or further contact people regarding, viruses associated with indigenous species of Aotearoa New Zealand. The cited literature within primary publications was investigated for further examples of published descriptions of relevant viruses. The National Center for Biotechnology Information (NCBI) Virus NCBI Virus (nih.gov, accessed 24 December 2021) was searched, filtered for the geographic region “New Zealand”, and the data were screened for accession with publications available and for indigenous species. Novel listings that described viruses of Aotearoa New Zealand indigenous species were included on the compiled list. Metadata for each virus species included host type, virus family/subfamily/genus (if known), virus identifier, indigenous host reported in association, known non-indigenous hosts, sample used for identification, location sampled, detection method, associated impact (if any), vector (if undertaken or known), management (if present), keywords used in search/method of search, year reported, and reference. 

### 2.2. Virus Detection Methods

Viruses were classified into one of three detection methods, termed as ‘symptom’, ‘targeted’, and ‘non-targeted’ identification. Viruses were placed under the ‘symptom’ detection method if the publication in which they were described mentioned that symptoms were observed on the host, and that triggered the investigation, and the potential use of other detection methods. Viruses identified via ‘targeted’ detection methods required no observation of symptoms, but required the use of methods that relied on either prior knowledge of the viruses present or a general examination of specific known viruses. Examples of targeted detection methods include polymerase chain reaction (PCR), reverse transcription polymerase chain reaction (RT-PCR), enzyme-linked immunosorbent assays (ELISA), immunodiffusion tests, and established plaque reduction tests for virus-specific neutralizing antibodies. Viruses identified via ‘non-targeted’ detection methods primarily involve broad-spectrum testing used without prior knowledge of any infecting virus. Examples of non-targeted detection methods include high-throughput sequencing applied to total RNA, and either RNA or DNA derived from partially purified virions or virus-enriched nucleic acid samples.

### 2.3. Designation of Virus Biostatus

For the purposes of this study, we defined the biostatus terms ‘endemic’, ‘indigenous’, and ‘exotic’ species based on their evolutionary place of origin using as biogeographical proxies: (i) the presence or absence of an association with an endemic, indigenous or exotic host; and (ii) the presence or absence within the ecosystem/country of interest (Table 1). An endemic species is one that evolved in and is naturally present in a single ecosystem/country. An indigenous species (a superset of an endemic species) is one that evolved in an ecosystem/country, but that can also naturally travel to other ecosystems/countries without human aid. An exotic species is one which has entered an ecosystem/country, with or without human aid, in which it did not evolve and was not previously present. 

An endemic virus was defined as one associated with an Aotearoa New Zealand endemic host type and present only in this country. There is potential for spill-over to occur to indigenous or exotic host species present within the country. 

An indigenous virus was defined as a virus that is reported associated with an Aotearoa New Zealand indigenous host, with the potential of travelling outside the country with that host. Indigenous viruses may have spilt-over into non-indigenous hosts. 

Exotic viruses were defined as those that were identified associated with hosts not endemic or indigenous to Aotearoa New Zealand, although these may also be present in Aotearoa New Zealand endemic or indigenous hosts following spill-over events. 

Undetermined viruses were defined if the indigenous Aotearoa New Zealand host species was known to travel outside the country’s borders; undetermined viruses may be indigenous or exotic, but not endemic.

### 2.4. Virus Taxonomic and Physical Metadata

Taxonomic and physical metadata (e.g., genome and virion size and composition) for each of the virus families in this study were collated from the websites https://talk.ictvonline.org/ and https://viralzone.expasy.org/ (accessed on 1 April–31 May 2022). 

## 3. Results

### 3.1. Discovery of Viruses Associated with Aotearoa New Zealand Indigenous Species Is Influenced by Discovery Method

The number of Aotearoa New Zealand indigenous viruses discovered and described in 64 published journal articles totals 701 (Figure 1, detailed in Appendix A). In total, across the entire 56 years of virus discovery in Aotearoa New Zealand indigenous species, 35 viruses were identified via symptom expression, 117 were from targeted detection, and 549 were from non-targeted detection. 

Our knowledge of the number of indigenous Aotearoa New Zealand viruses has increased relatively slowly over time, until around 2010, wherein an exponential increase occurred. Seventeen viruses were discovered prior to 2000 (an average of 0.5 viruses per year), 21 viruses were discovered between 2000 and 2009 inclusive (an average of 2.1 viruses per year), and 663 viruses were discovered between 2010 and 2022 (an average of 52.8 viruses per year). Viruses have been discovered on the basis of symptoms for all 56 years (an average of 0.6 viruses described per year), by targeted methods for 42 years (an average of 2.1 viruses described per year), and by non-targeted since 2010 (an average of 45.8 viruses described per year). Non-targeted methods have given rise to a 22-fold increase in virus discovery rate over targeted methods and 73-fold over symptom-based methods.

Of the techniques used in Aotearoa New Zealand to report viruses, non-targeted approaches have identified the highest number of unique viruses (549, Figure 1). The application of non-targeted approaches has allowed for an exponential increase in virus reporting from 2010 that has not yet plateaued. This equated to an average reported virus discovery rate of 45.75 viruses per year by non-targeted methods alone compared with 0.93 per year by any method prior to 2010. From this dataset, the average per annum reported virus discovery rate since 1966 was lowest using symptom observations as the initial detection method (average rate of 0.63, i.e., 35 viruses/56 years), middle by targeted methods (average rate of 2.1, i.e., 117 viruses/56 years), and highest by non-targeted methods (average rate of 9.8, i.e., 549 viruses/56 years). 

There have been 64 total publications on viruses discovered associated with Aotearoa New Zealand indigenous species over the past 56 years (an average of between one to two publications per year, describing on average 11 viruses per publication) (Figure 1). Of these, 23 used symptoms to initially detect virus presence (an average of less than one publication per year describing one to two viruses per publication), 16 used targeted methods (an average of less than one publication per year describing over seven viruses per publication), and 25 used non-targeted methods (an average of less than two publications per year describing 22 viruses per publication). Across time, there were 12 publications prior to the year 2000 (an average of less than one publication per year, describing less than two viruses per publication), 11 publications between 2000 and 2009 (an average of one to two publications per year, describing less than two viruses per publication), and 41 publications between 2010 and 2022 (an average of three to four publications per year, describing 16 viruses per publication). 

The differences in viral discovery rate are dependent on both the method of detection and host type (Figure 2). Non-targeted approaches have resulted in the highest number of viruses reported in individual host types, with insects and birds representing 280 (40%) and 189 (27%) viruses reported, respectively. Non-targeted methods were used to discover and report viruses associated with the most diverse number of different hosts, whereby the eight different Aotearoa New Zealand indigenous host types were associated with at least one reported virus. The identification of viruses using symptoms for initial detection or targeted methods resulted in viruses associated with four or three host types, respectively. Targeted approaches were used to identify and report a relatively high number of viruses identified in plants compared with other host types. Thirty-five of the viruses reported associated with Aotearoa New Zealand indigenous species were initially identified by observed symptoms expressed by the plant (24), bird (7), fish (3), or insect hosts (Figure 2, Appendix A). For all viruses associated with visual symptoms, except three coronaviruses identified in fish, the initial observation was followed by a targeted approach to identify the disease-causing agent. Regardless of virus discovery method, the three host types associated with the highest number of reported viruses are insects (280), birds (189), and plants (126). 

### 3.2. To Date, Vast Viral Diversity Has Been Described Associated with Aotearoa New Zealand Indigenous Macro- But Not Micro-Flora and Fauna 

The indigenous Aotearoa New Zealand host groups associated with reported viruses are plants, insects, birds, bats, fish, seals, mollusks, and starfish (*Asteroidea*) (Figure 2, Appendix A). Thus far, no indigenous viruses have been reported associated with indigenous archaea, oomycetes, fungi, or bacteria.

#### 3.2.1. Plants

To date, 126 viruses belonging to 12 families have been reported to be associated with 43 Aotearoa New Zealand indigenous plant hosts, belonging to 16 families. These viruses have been identified from leaves, tillers, and flower petals, both through classical plant virology methods (including the observation of symptoms, transmission experiments including the mechanical inoculation of indicator plants, and the introduction of indicator plants with suspected viral inoculum) and by enzymatic methods that predominantly target specific viruses (such as PCR, RT-PCR, ELISA, and a range of sequencing methods) (Figure 2, Appendix A).

#### 3.2.2. Insects

To date, 280 viruses from 15 families have been reported to be associated with indigenous insects (Figure 2, Appendix A). Viruses have been identified via random sampling, DNA isolation then Illumina sequencing, electron microscopy, and a biological assay, the plaque reduction test. All insect-associated viruses identified may be endemic, as they have thus far been detected only in Aotearoa New Zealand indigenous species. However, there has been no extensive or targeted research undertaken to determine their endemism.

#### 3.2.3. Birds

To date, 189 viruses from 11 families are reported to be associated with indigenous birds (Figure 2, Appendix A). Although classified into families, these viruses are largely unnamed. Those that have been named include beak and feather disease virus (Circovirus), reported to be associated with indigenous hosts *Cyanoramphus novaezelandiae* (red-fronted parakeets/kākāriki), *Cyanoramphus auriceps* (yellow-crowned parakeet/kākāriki) and *Cyanoramphus unicolor* (antipodes island parakeet/pākāriki) [16,17,18], and the Whataroa virus (Alphavirus), reported in *Zosterops lateralis* (silvereye or waxeye/tauhou). The identified viruses have been reported in the families *Circoviridae*, *Flaviviridae*, *Poxviridae*, and *Togaviridae*, mainly through molecular techniques, including in situ hybridization, PCR, as well as various DNA sequencing methods. The non-molecular methods used were the plaque reduction test and ELISA, and the visualization of symptoms on sick birds was an indication of potential virus infection. 

#### 3.2.4. Bats

Three species of endemic bats are Aotearoa New Zealand’s only indigenous land mammals [19]. The bat-associated viruses were identified through molecular means, from random sampling of guano, DNA, or RNA extraction, before Illumina sequencing and bioinformatics analyses. In the *Mystacina tuberculata* (lesser short-tailed bat/ pekapeka-tou-poto), 13 viruses were identified from five families, including *Caliciviridae*, *Coronaviridae*, *Hepeviridae*, *Papillomaviridae*, and *Polyomaviridae* (Figure 2, Appendix A). 

#### 3.2.5. Marine Life

The final indigenous species with reported viruses associated include those reported in marine environments. These include mollusks, starfish, fish, and seals (Figure 2, Appendix A). Mollusks have 72 viruses identified, belonging to four families. These viruses were all identified through non-targeted methods: random sampling, DNA extraction, rolling circular/circle amplification (RCA), and then either Sanger or Illumina sequencing. Asteroids, also known as starfish or sea stars, had four viruses belonging to one family reported in two indigenous sea stars. These viruses are from the family *Parvoviridae*, and were reported in the species *Coscinasterias muricata* (eleven armed sea star/patakaroa) and *Patiriella regularis* (New Zealand common cushion star), and are thus far unnamed. After random sampling of the asteroid body wall and RNA extraction, Illumina sequencing was performed to identify and sequence viral presence [20]. There were 15 viruses, belonging to nine different families, reported in fish. These were identified through Illumina sequencing following symptom observation or random sampling. Lastly, two viruses belonging to *Circoviridae* and *Gemonoviridae* were reported to be associated with seals. These were identified through random sampling before being sequenced by Illumina sequencing and primer walking [21].

### 3.3. The Majority of Viruses Associated with Indigenous Species in Aotearoa New Zealand Are Themselves Classified as Indigenous 

The vast majority of reported viruses that were discovered in association with an indigenous host within Aotearoa New Zealand are themselves classified as indigenous (Figure 3). In particular, from 2010 onwards, there was an exponential increase in the number of Aotearoa New Zealand indigenous viruses reported, in contrast to the near-static discovery and reporting of exotic and undetermined viruses. This exponential increase correlates with the use of non-targeted detection methods (Figure 3). 

Compared with the non-targeted approaches, few viruses (many of which are exotic) were discovered via symptom observation or using targeted approaches. Of the 35 viruses reported following detection via the symptom expression of their host, two thirds (23) are exotic to New Zealand, four are indigenous, and eight are undetermined. Viruses discovered by way of targeted approaches totaled 117 (44 exotic, 65 indigenous, and 8 undetermined). Regardless of their discovery method, all plant viruses except one are classified as exotic to New Zealand.

To the best of our knowledge, only one virus (Whataroa virus) was discovered and reported to be associated with an Aotearoa New Zealand indigenous (or endemic) species and then secondarily detected and reported in an exotic host species (Appendix A). All 104 exotic viruses catalogued in this report are examples of viruses that have spilt over into Aotearoa New Zealand indigenous (or potentially endemic) host species (Appendix A). 

### 3.4. Detection Ease and Sum of Viruses Described Associated with Aotearoa New Zealand Indigenous Hosts Varies between Virus and Host Types

The investigation into the connection of virus family to the number of viruses reported was conducted by graphing viruses dependent on (1) their detection method, (2) DNA or RNA genome, and (3) the virus family (Figure 4). This showed that there were similar numbers of reported DNA and RNA viruses identified using symptomatic and targeted approaches, and RNA viruses using non-targeted approaches, but there was a large increase (~450 viruses) for DNA viruses identified using non-targeted approaches. The two families with the highest numbers of reported viruses were unclassified circular, Rep-encoding ssDNA (CRESS DNA), and *Microviridae*. These, in turn, were associated with bird and insect host types. To investigate whether physical factors of specific virus taxa may predicate virus discovery, we ranked the reported virus number with genome and virion type, size, and composition. Appendix A shows a correlation between the number of small DNA viruses reported, including unclassified CRESS DNA, *Microviridae*, and *Circoviridae*, their genome size (which is relatively small compared with those of other families, ~2 kb versus 5–20 kb for other viruses), and the high number of viruses detected. All these families contain small single-stranded monopartite DNA, where the particles are non-enveloped, and the virion is 15–30 nm in diameter, with icosahedral T = 1 symmetry. 

## 4. Discussion 

### 4.1. Indigenous and Endemic Viruses of Aotearoa New Zealand 

Within Aotearoa New Zealand, endemic and indigenous species hold special cultural status [22,23]. There is an innate need to talk with the indigenous people of the land (mana whenua) when working with indigenous or endemic species of Aotearoa New Zealand, as these people and species have shared history and presence (whakapapa). Such conversations may result in a name gifted in the Māori language (te reo Māori), e.g., of a potential virus host, such as a whale or of a virus, and implications for the management of the species [24,25,26,27].

Here, we defined the terms to categorize whether a virus is ‘endemic’, ‘indigenous’, ‘exotic’, or ‘undetermined’; however, future categorization may change as new data are generated and our understanding of the distribution and host range of these viruses expands. In this study, ‘indigenous’ and ‘endemic’ mean naturally occurring in the location (with the superset term indigenous used predominantly, as the endemic state may be uncertain); in contrast, exotic species are recent arrivals (e.g., by human introduction to the location). Therefore, for both indigenous and exotic species to be classified, their distribution and mode of arrival must be understood. Only large datasets can trace back the origin of a virus, as exampled for some plant viruses [28,29,30].

Intriguingly, to date, no viruses have been reported to be associated with Aotearoa New Zealand indigenous archaea, oomycetes, fungi, or bacteria, perhaps because they and their infection status are not easily visualized. The categorization of ‘indigenous’ and ‘endemic’ fungi’ has begun recently for the International Collection of Microorganisms (ICMP, living cultures of plant-associated bacteria, and of fungi from New Zealand and the South Pacific) (Landcare Research, 2013–2022). ‘Indigenous’ also has social, cultural, and scientific context shifts, as it is heavily influenced by the wider environment and understanding [31]. For instance, the bird *Zosterops lateralis* (silvereye or wax-eye/tauhou, meaning ‘new arrival’ in te reo Māori) is considered an indigenous species despite reportedly only arriving in Aotearoa New Zealand in the 1800s [32,33,34]. Although this study has distinguished viruses associated with indigenous and exotic species, some changes to virus designation may arise if new research reveals an alternative arrival history of a host. 

The breadth of host range affects endemism or indigeneity. Viruses rely on their hosts for replication, so single-host (specialist) viruses are endemic wherever their host is endemic. However, it is very difficult to know if a virus is specialist or generalist based on its sequence alone (and it can change) and often, the data of viruses detected from indigenous species are scarce (often one sole sequence). On the other hand, broad host range (generalist) viruses identified associated with Aotearoa New Zealand indigenous species are mostly classified as exotic, being introduced with exotic species and spilled over into the natural fauna or flora within the new country. To date, we have only identified one indigenous virus, Whataroa virus, that has a broad host range, though more indigenous viruses with a broad host range may be described in the future (Table 1).

The historic isolation of Aotearoa New Zealand may explain how the majority of viruses detected in its endemic species have been classified as indigenous. To date, we have counted 701 viruses described in published journal articles from NZ endemic hosts, and of these, we listed 553 as indigenous (Figure 1), and many of these indigenous viruses are likely also to be endemic. Aotearoa New Zealand is an island nation that has no land borders with other countries; it is part of the continent Zealandia that split from its supercontinent Gondwanaland about 80 million years ago [35,36]. This physical isolation has resulted in unique evolutionary trajectories amongst many of the indigenous lineages, including gigantism. Until settlement by Māori, then Europeans, and the subsequent introductions of multiple exotic species and associated viruses, only a few migratory species were part of the unique array of Aotearoa New Zealand indigenous species, which limited the transmission routes of native viruses beyond the country. This predominantly closed ecosystem has probably led to the coevolution of viruses and their hosts, and the disproportionately high endemism of viruses. 

Given this isolated environment, viruses only reported in endemic species were classified as endemic, e.g., three of the identified endemic bird viruses not reported outside Aotearoa New Zealand. These include the Rowi kiwi circovirus-like virus in *Apteryx rowi* (okarito brown kiwi/rowi), and two poxviruses in *Megadyptes antipodes* (yellow-eyed penguin/hoiho). Likewise, bats are the only indigenous land mammals present in Aotearoa, comprising two endemic species, *Mystacina tuberculata* (lesser short-tailed bat/pekapeka) and *Chalinolobus tuberculatus* (New Zealand long-tailed bat/pekapeka-tou-roa). Their viruses were considered endemic to Aotearoa New Zealand. Migratory birds are the only indigenous (land-based) species naturally reaching other lands while migrating. As a result, it is difficult to classify a circovirus reported in *Larus dominicanus* (southern black-backed gull/karoro) [37,38]. This species of bird is known to travel to Australia and South America. Without a large amount of sequencing data of this virus from all of these countries, we cannot know where the virus originally came from, illustrating the difficulty of establishing endemism of migrating bird viruses. 

### 4.2. Implications of New Virus Discovery in Aotearoa New Zealand 

The under-identification of viruses worldwide confounds individual countries’ or regions’ biosecurity and research containment processes [39]. Newly described Aotearoa New Zealand species, including viruses, must be confirmed as legally present in the country prior to 29 July 1998 or they are determined as ‘new’ to the country and research on them is consequently restricted to physical containment facilities. New Zealand’s Hazardous Substances and New Organisms Act 1996 (as at 28 October 2021) defines a ‘new organism’ as ‘*an organism belonging to a species that was not present in New Zealand immediately before 29 July 1998’* (section 2A)(2)(a) [40,41]. However, an organism is not a new organism if ‘*the new organism was deemed to be a new organism under section 255 and other organisms of the same taxonomic classification were lawfully present in New Zealand before the commencement of that section and in a place that was not registered as a circus or zoo under the Zoological Gardens Regulations 1977*.’ (Section 2A)(2)(c). Though appropriate for larger, easily detected organisms, this definition can be difficult to prove for microorganisms, including viruses. Therefore, to avoid classification as ‘new’, a virus isolated from an Aotearoa New Zealand indigenous species must either be isolated from a sample collected prior to that date, or be demonstrated as ubiquitous in nature through multiple isolations across the country suggesting that there have been multiple generations of the same organism present. Physical containment is required for organisms that are ‘new’, which forms a costly hurdle to any research (beyond identification) on microorganisms in indigenous species.

### 4.3. Virus Abundances within Aotearoa New Zealand 

This report has investigated, from scientific publications, the viral abundances associated with Aotearoa New Zealand indigenous hosts according to four interlinked factors: host type; detection method used; virus status as ‘indigenous’ or not; and virus family. In short, viral identification was highest in the plant, bird, and insect hosts; the highest detection was obtained using non-targeted detection methods; viruses classified as indigenous were more prevalent than exotic (on indigenous hosts); finally, the virus family *Microviridae* and the unclassified CRESS DNA were overrepresented in the list of described viruses from Aotearoa New Zealand indigenous hosts.

Multiple causes, such as host sampling bias, identification method, and virus family, have influenced this ‘snap-shot’ of Aotearoa New Zealand indigenous viromes. Despite these biases, our analyses suggest the viruses infecting indigenous species in Aotearoa New Zealand are largely indigenous themselves.

Host sampling bias is multi-factored, including the presence of easily identifiable symptoms (especially on birds and plants); specific interest in particular host types; the prevalence of the host; and the involvement of specific scientists/research teams and their respective research topics. The consequences of these viruses on host health and fitness, however, requires further investigation. For example, we have catalogued viruses associated with several host types of higher tropic hierarchies, such as insects, birds, and bats, that may have accumulated in the guts of these hosts and, therefore, may be only associated (i.e., not biologically infect the associated ‘host’). It is important to note that the organism where a virus is discovered is not necessarily the host of the virus, and biological research needs to clarify these intimate relationships.

Most indigenous viruses were discovered by non-targeted approaches. These newly detected viruses were previously overlooked (no symptoms and/or no value perceived of the host). Since the rate of non-targeted virus discovery does not appear to have reached a plateau, there are probably many other undiscovered Aotearoa New Zealand indigenous viruses. Non-targeted, high-throughput sequencing has dramatically increased viral discovery by ~22- and ~73-fold over targeted and symptom-based discovery methods, respectively. 

Virus family relates strongly with the number of reported viruses. The highest numbers of viruses reported are from two virus groups: unclassified CRESS DNA and in the family *Microviridae*. This is due to the type of detection methods used, as these share virus genome structures. RCA specifically targets circular DNA, and was used prior, or in association with, high-throughput sequencing to discover most of the reported unclassified CRESS DNA viruses. This virus group primarily comprised Aotearoa New Zealand indigenous viruses, and was predominantly associated with insects. Enrichment methods such as RCA allow for the relatively quick and easy processing of DNA prior to sequencing and can be applied to large numbers of samples, yielding numerous new-to-science virus discoveries (such as unclassified CRESS DNA and *Microviridae*). 

### 4.4. Impacts on Virus Management 

The availability of non-targeted detection tools has expanded the field of virology research to a wide range of species and ecosystems, particularly indigenous species that are often asymptomatic [42,43]. However, the affordable acquisition of a large volume of sequence data leaves a large gap in biological information that is expensive to fill [5]. The dearth of biological knowledge makes it difficult to evaluate the implications to biosecurity, regulatory, and management decisions from non-targeted results alone [5,39,44].

Viruses with efficient vectors and broad host range are most likely to be associated with an increased risk of new disease outbreaks [45,46]. Changes to land use are increasingly altering the interface between the crop and indigenous plant communities, as well as between wild animals and farm animals. Here, we did not observe obvious spill-over of viruses from an indigenous species to an exotic crop plant, which may be of concern to primary industries. Of more concern to indigenous species, we observed the opposite, with spill-over of plant viruses from exotic crop plants to indigenous plants, which often resulted in symptoms, as demonstrated by the high number of exotic plant viruses identified by symptoms or by targeted methods (Figure 2). For example, barley yellow dwarf virus has been observed in many indigenous hosts, including, but not exclusive to, *Microlaena stipoides* (meadow rice grass/pātītī), *Hierochloe redolens* (holy grass/karetu), and *Poa cita* (silver tussock/wiwi). The capability of viruses to infect hosts across kingdoms is commonly reported and will likely feature in the ecology of the Aotearoa New Zealand indigenous virome [47]. The karaka Ōkahu purepure virus, first identified in an indigenous plant species, does not systemically infect the plant; the natural host of the virus is more likely to be the endemic karaka gall mite [26].

### 4.5. Current and Future Methods of Detection

In order to study the motivations for virus discovery, we have classified them in three categories: symptoms, targeted, and non-targeted. An additional approach, which has not yet been used on our shores, is that of detection made *a posteriori* through the mining of existing databases. The amount of sequence archives publicly available has the potential to give rise to an unprecedented wave of new virus identifications. Mining publicly available sequence databases is a practice that has been used widely to detect viruses [30,48,49], but has not yet been applied to Aotearoa New Zealand-derived sequences. Serratus, a recent open-science viral discovery platform, identified 10^5^ novel RNA viruses from the publicly available Sequence Read Archives (SRAs), based on the RdRp motif [50]. To date, the thousand-plus RNA-based SRAs from Aotearoa New Zealand samples are mostly from exotic species, i.e., crops and livestock (1120 SRAs detected when selecting New Zealand on the ‘The Planetary RNA Virome’ from Serratus, https://serratus.io/geo accessed on the 7 July 2022). However, some Aotearoa New Zealand indigenous species SRAs are available. Sequencing data from the studies of the control of flowering in *Celmisia lyallii* (*Asteraceae*) PRJNA576138) [51] harbor viral RdRp motifs from putative novel viruses within the families *Solemoviridae* and *Partitiviridae*, as identified with the tool search of Serratus PalmID. Similarly, several putative viral RdRp motifs can be found in SRAs from the costal spider *Desis marina* (SRR12968639) [52] or from the New Zealand giant collembola (*Holacanthella duospinosa*) transcriptome (SRR5626542) [53]. There is no doubt that a comprehensive review of the SRA-based data for Aotearoa New Zealand indigenous species would significantly alter the catalogued data presented here, i.e., only those viruses associated with Aotearoa New Zealand indigenous species that have been reported in peer-reviewed journals. Such a study would shed new light on the diversity and potential uniqueness of the Aotearoa New Zealand virome. Currently, it is estimated that only 0.1% of all viruses have been described worldwide [50]. With the continued addition of viral sequences, the increased information on their genome and calculated evolution, as well as their distribution and host range, will provide valuable information to trace their origins and affect the designation of exotic, indigenous, and endemic to specific countries or ecosystems. 

## 5. Conclusions

This is the first catalogue of viruses associated with indigenous species of a country. The catalogue does not include all viruses reported within Aotearoa New Zealand, but has prioritized indigenous host species, as these hold significant cultural, social, and economic importance. High-throughput sequencing methods, often preceded by an enrichment process, has resulted in an exponential number of reported new-to-science viruses that are probably indigenous to the country. However, to this date, several biases have been highlighted, mainly caused by the enrichment tools selected for detection and the hosts studied. The pace of virus detection and reporting promises to shape a broader and more accurate indigenous virome in the future. This knowledge is needed for cultural, biosafety, and research purposes. For instance, we need to study the dynamics of viruses between endemic and exotic ecosystems to better protect threatened endemic species. It is also essential to explore the evolution of viruses and to understand their involvement in shaping the unique ecosystem of Aotearoa New Zealand.

## Figures and Tables

**Figure 1 viruses-14-02477-f001:**
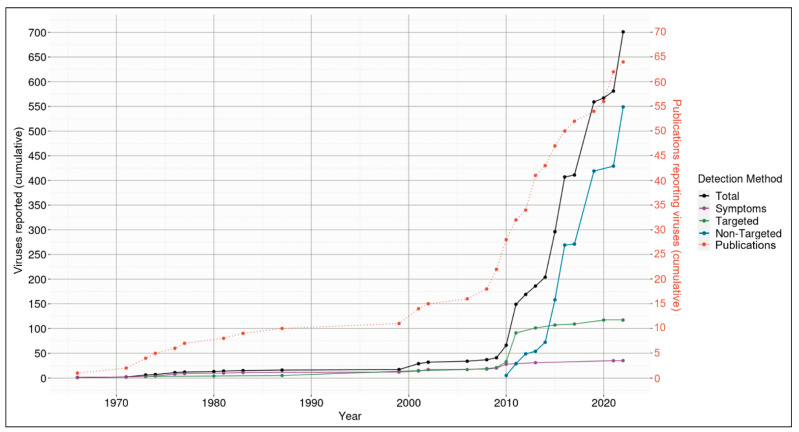
The cumulative viruses reported across time from 1966 to February 2022, with a specific focus on the detection method used for identification, and the number of publications in Aotearoa New Zealand indigenous hosts. ‘Total’ refers to all viruses discovered in Aotearoa New Zealand, independent of detection method. ‘Symptoms’ refers to viruses detected with symptom observation. ‘Targeted’ refers to viruses identified via targeted approaches for known viruses. ‘Non-targeted’ refers to broad-spectrum viral identification. The secondary axis in orange ‘Publications reporting viruses (cumulative)’, associated with the dotted line, refers to the number of publications reporting the viruses.

**Figure 2 viruses-14-02477-f002:**
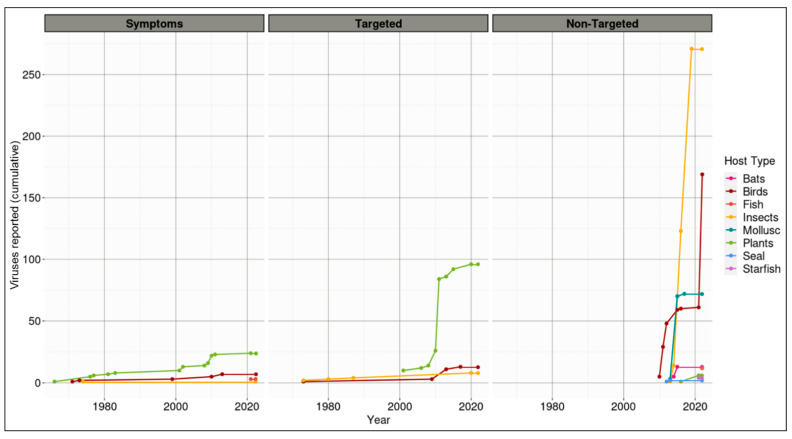
Total accumulated viruses reported (prior to 28 February 2022) associated with an Aotearoa New Zealand indigenous host, dependent on host type and detection method.

**Figure 3 viruses-14-02477-f003:**
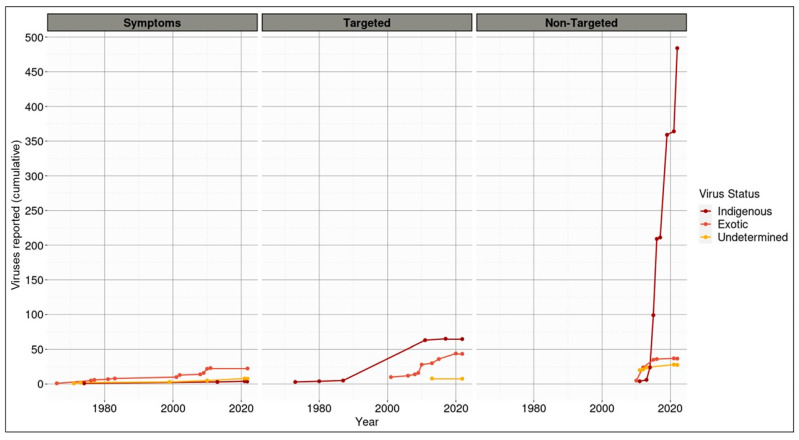
Total number of reported viruses associated with Aotearoa New Zealand indigenous hosts as of February 2022, compared based on virus status and detection method. Virus status is described in Section 2.4.

**Figure 4 viruses-14-02477-f004:**
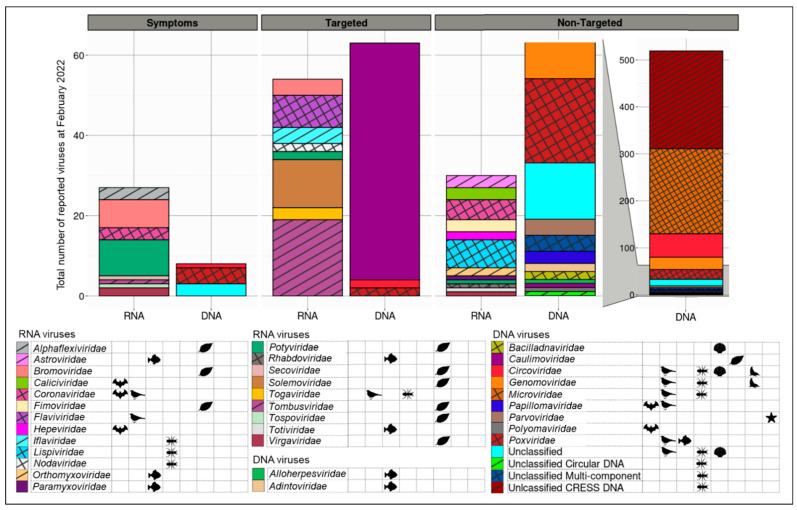
Total number of reported viruses associated with Aotearoa New Zealand indigenous hosts as of February 2022, compared based on identification method and viral family. Identification method refers broadly to how a virus was discovered in a New Zealand indigenous host. ‘Symptoms’ refers to viruses detected by symptom observation. ‘Targeted’ refers to viruses identified via targeted approaches with known viruses. ‘Non-targeted’ refers to broad-spectrum viral identification (for DNA viruses there are two columns, as indicated by the grey shading; the left column is a subset of the right one, which has its own scale). Viral families are split into RNA and DNA, and the hosts in which these viruses have been reported during the course of this meta-analysis have been identified via associated symbols: bats

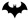
, birds

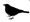
, fish

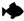
, insects

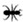
, mollusks

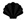
, plants

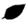
, seals

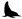
, and starfish

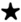
.

**Table 1 viruses-14-02477-t001:** Virus biostatus designations used in this study. Designations change as more information on virus evolutionary history, hosts, and biogeography are revealed.

Virus Biostatus Designation	Associated Host			Described Outside Country	Potential Spill-Over Events and Other Comments
	Endemic	Indigenous	Exotic		
Endemic	Y	N	N	N	
Endemic or indigenous	Y	Y	N	N	
Endemic or indigenous	Y	Y	Y	N	Spilt to exotic hosts
Indigenous	N	Y	N	N	
Indigenous	N	Y	N	Y	
Indigenous	N	Y	Y	N	Spilt to exotic hosts
Exotic	N	N	Y	Y	Not catalogued in this study
Exotic	Y	N	Y	Y	Spilt to endemic hosts
Exotic	N	Y	Y	Y	Spilt to indigenous hosts
Exotic	Y	Y	Y	Y	Spilt to endemic and indigenous hosts
Undetermined	N	Y	N	Y	Possibly indigenous, but found outside of NZ. May also spill to endemic and/or exotic hosts.

## Data Availability

Not applicable.

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
