# Peer review of "A National Catalogue of Viruses Associated with Indigenous Species Reveals High-Throughput Sequencing as a Driver of Indigenous Virus Discovery"

_viruses, 2022, doi:10.3390/v14112477_

Round 1

Reviewer 1 Report

This Aotearoa New Zealand virus/host data analysis provides novel and interesting insights on several fronts as clearly reported and critically discussed by the authors. The article was a pleasure to read. There are only some minor issues that should be addressed:

Luteoviridae family abolished in 2020, genus Polerovirus now classified in family Solemoviridae. (Table S1_plant hosts)

Last para on page 5: The averaging of publication numbers and reported viruses per year or per publication does not make much sense beyond mathematical interest when for example an average of 0.4 publications and 1.4 viruses are mentioned. Maybe say “less than 1 publication and less than 2 viruses”?

L229: “and mechanical inoculation of indicator plants”

L303 (Fig. 3 legend): delete “used for isolation” since using at least the non-targeted detection methods, the viruses would not have been isolated.

L312: define “CRESS” DNA

Author Response

Reviewer 1

  • Luteoviridae family abolished in 2020, genus Polerovirusnow classified in family Solemoviridae. (Table S1_plant hosts)

Rows 564-570 inclusive in Supplementary Table 1 have changed Luteoviridae to Solemoviridae as per the ICTV Virus Taxonomy: 2021 Release EC 53, Online, July 2021, Email ratification March 2022 (MSL #37).

  • Last para on page 5: The averaging of publication numbers and reported viruses per year or per publication does not make much sense beyond mathematical interest when for example an average of 0.4 publications and 1.4 viruses are mentioned. Maybe say “less than 1 publication and less than 2 viruses”?

Changed all reference to numbers as suggested.

  • L229: “and mechanical inoculation of indicator plants”

Changed as suggested

  • L303 (Fig. 3 legend): delete “used for isolation” since using at least the non-targeted detection methods, the viruses would not have been isolated.

Changed as suggested

  • L312: define “CRESS” DNA

Included circular, Rep-encoding ssDNA (CRESS DNA)

Reviewer 2 Report

The present study summarizes current knowledge on extant virome in Aotearoa New Zealand, underlining the determining role HTS technology played in virus discoveries, especially regarding indigenous viruses. The paper could be also considered a very informative and well-written review on the issue, while interesting scientific questions are arise, considering

- l. 47: Recent reviews have summarized novel virus discoveries thanks to HTS, which come in accordance to the results of the present study. Representative ones from Europe are for example

a. Rumbou, A.; Vainio, E.J.; Buttner, C. Towards the Forest Virome: High-Throughput Sequencing Drastically Expands OurUnderstanding on Virosphere in Temperate Forest Ecosystems. Microorganisms 2021, 9, 1730. [CrossRef]202.

b. Maclot, F.; Candresse, T.; Filloux, D.; Malmstrom, C.M.; Roumagnac, P.; van der Vlugt, R.; Massart, S. Illuminating an EcologicalBlackbox: Using High Throughput Sequencing to Characterize the Plant Virome Across Scales. Front. Microbiol. 2020, 11, 578064.
c. Hou, W.; Li, S.; Massart, S. Is There a “Biological Desert” With the Discovery of New Plant Viruses? A Retrospective Analysis for New Fruit Tree Viruses. Front. Microbiol. 2020, 11, 592816.

- Figure 3. Better figure quality could be provided.

- l. 33-339: The meaning of this introductory paragraph is not really clear to me, probably due to missing background knowledge. What does for example mean “a name gifted in the MaÌ„ori language” (?). For what reason there is a specific need to talk with the indigenous people about endemic species? What information would the local people deliver and what implications could that have to the virus research? Could you please be more explanatory on this issue?

Author Response

Reviewer 2

  • 47: Recent reviews have summarized novel virus discoveries thanks to HTS, which come in accordance to the results of the present study. Representative ones from Europe are for example

  • Rumbou, A.; Vainio, E.J.; Buttner, C. Towards the Forest Virome: High-Throughput Sequencing Drastically Expands OurUnderstanding on Virosphere in Temperate Forest Ecosystems. Microorganisms 2021, 9, 1730. [CrossRef]202.

  • Maclot, F.; Candresse, T.; Filloux, D.; Malmstrom, C.M.; Roumagnac, P.; van der Vlugt, R.; Massart, S. Illuminating an EcologicalBlackbox: Using High Throughput Sequencing to Characterize the Plant Virome Across Scales. Front. Microbiol. 2020, 11, 578064.

  • Hou, W.; Li, S.; Massart, S. Is There a “Biological Desert” With the Discovery of New Plant Viruses? A Retrospective Analysis for New Fruit Tree Viruses. Front. Microbiol. 2020, 11, 592816.

Included within text and reference list

  • Figure 3. Better figure quality could be provided.

Higher quality images are available upon request by the Viruses editorial team and will be uploaded by the author if possible during the revision process

  • 33-339: The meaning of this introductory paragraph is not really clear to me, probably due to missing background knowledge. What does for example mean “a name gifted in the MaÌ„ori language” (?). For what reason there is a specific need to talk with the indigenous people about endemic species? What information would the local people deliver and what implications could that have to the virus research? Could you please be more explanatory on this issue?

Clarified with the addition of two references:

Waitangi Tribunal. 2011. Ko Aotearoa Tēnei: Report on the Wai 262 Claim. https://www.waitangitribunal.govt.nz/news/ko-aotearoa-tenei-report-on-the-wai-262-claim-released (Accessed 3 November 2022)

Ngaaho 1993. The Mataatua Declaration on Cultural and Intellectual Property Rights of Indigenous Peoples https://ngaaho.maori.nz/cms/resources/mataatua.pdf (Accessed 3 November 2022)